# The Effect of Routine High-Soy-Protein Soymilk Intake on the Skeletal Muscle Index (SMI) in Japanese Pre-Frail Elderly Individuals with an Ordinary Life: A Post Hoc Analysis of a Randomized Controlled Trial

**DOI:** 10.3390/nu17243900

**Published:** 2025-12-13

**Authors:** Daigo Sakamoto, Yuji Terashima, Makoto Sugawara, Ryoichi Unno, Tomoyuki Watanabe, Tomoko Uno, Mitsuo Maruyama

**Affiliations:** 1Research and Development Division, MARUSAN-AI Co., Ltd., Okazaki 444-2193, Aichi, Japan; daigo.sakamoto@marusanai.co.jp (D.S.); makoto.sugawara@marusanai.co.jp (M.S.); ryoichiunno@gmail.com (R.U.); 2Department of Health and Nutritional Sciences, Faculty of Health Sciences, Aichi Gakuin University, Nisshin 470-0131, Aichi, Japan; wtomo709@dpc.agu.ac.jp (T.W.); tomoko26@dpc.agu.ac.jp (T.U.); 3Geroscience Research Center, Research Institute, National Center for Geriatrics and Gerontology, Obu 474-8511, Aichi, Japan; 4Department of Aging Research, Nagoya University Graduate School of Medicine, Nagoya 466-8550, Aichi, Japan

**Keywords:** frailty, soy protein, soymilk, nutritional intervention, appendicular skeletal muscle mass, skeletal muscle index

## Abstract

Background/Objectives: The primary outcome of a 12-week randomized controlled trial studying the effects of continuous high-soy-protein (HSP) soymilk consumption in Japanese frail and pre-frail elderly individuals has previously been reported. The authors of this post hoc study aimed to conduct a statistical analysis focusing on changes in the skeletal muscle index (SMI) and evaluate the effect of HSP soymilk intake on appendicular skeletal muscle mass (ASM). Methods: A full analysis of all subjects (*n* = 68, HSP group: *n* = 34, control group: *n* = 34) and a subgroup analysis of only pre-frail elderly individuals (*n* = 58, HSP group: *n* = 29, control group: *n* = 29) were performed. The following nine subgroup analyses were conducted: frailty phenotype, basal metabolic rate, walking speed, step counts, total energy expenditure (TEE), total energy intake (TEI), estimated protein intake, age, and sex. Results: In the overall analysis (primary combined cohort), the SMI showed no significant change between the HSP and control soymilk groups (*p* = 0.18); however, a significant difference in the change in the SMI between the HSP and control groups was revealed for pre-frail elderly subjects (mean difference in SMI change: 0.08 kg/m^2^ [95% CI 0.004, 0.15], *p* = 0.044). Furthermore, in four distinct subgroups restricted to the pre-frail elderly individuals—step counts (≥5000 steps/day), TEE (≥30 kcal/kg BW/day), TEI (≥30 kcal/kg BW/day), and male sex—the HSP group also showed significant differences in SMI change. Conclusions: In this study, no significant improvement in the SMI was observed across all subjects (frail and pre-frail elderly individuals); however, an exploratory subgroup analysis suggested that continued daily HSP soymilk intake was associated with potential benefits in pre-frail elderly individuals with high physical activity and energy intake levels. These findings are preliminary and require further research.

## 1. Introduction

As a primary cause of physical frailty, sarcopenia is characterized by progressive skeletal muscle mass and strength loss. In particular, appendicular skeletal muscle mass (ASM), which reflects the limb musculature, plays a vital role in mobility and daily functioning. Nutritional strategies aimed at preserving or enhancing ASM are therefore essential in mitigating frailty-related risks; among these strategies, adequate protein intake has been consistently highlighted as a cornerstone of frailty prevention.

The Health, Aging, and Body Composition Study (The Health ABC Study) was a prospective cohort study (U.S. older adults: 70–79 years old; 3075 participants) demonstrating that higher protein intake among older adults was associated with better muscle mass maintenance and a lower risk of functional decline [1]. Recently, in a meta-analysis of 30 randomized controlled trials comparing plant and animal proteins’ effects on muscle health, Reid-McCann et al. reported that soy protein had a similar effect on muscle mass compared to dairy protein. Similarly, no significant differences were found between plant and animal proteins for muscle strength and physical performance [2].

Previous studies have reported that soy protein intake is effective for maintaining muscle mass and strength across various physical activity levels, with particularly pronounced effects in individuals with regular exercise habits [3]. Recent research indicates that soy protein offers benefits for muscle adaptation, antioxidant status, and hormonal responses, contributing to improved exercise performance [4]. Furthermore, plant-based proteins, including soy, have been reported to promote recovery from muscle damage and contribute to reduced inflammatory markers [5].

Furthermore, the results of a randomized controlled trial with soy-based protein suggested that nutritional intervention alone is not sufficient for frail elderly individuals and that combining it with exercise intervention is beneficial [6], while the authors of another study noted that soy protein supplement intake improves exercise function in elderly individuals with sarcopenia [7]. These reports suggest that physical frailty caused by decreased muscle mass in our extremities and physical function can be prevented by appropriate protein intake and regular exercise.

Our previous randomized controlled trial was conducted to examine the impact of a high-soy-protein (HSP) soymilk intervention on frailty, especially in Japanese pre-frail or frail elderly individuals aged 65 to 83 years, and the exploratory subgroup analysis results supported the possibility that the continued intake of soymilk with HSP content might improve the walking speed of elderly individuals with an ordinary speed of 1.0 m/s or higher or who usually reached 5000 or more steps per day [8]. However, the influence of continuous soy protein intake on ASM in frail or pre-frail elderly individuals remains to be sufficiently elucidated.

Hence, as a secondary analysis of our previous study [8], in this study, we conducted further statistical analysis, focusing on changes in the skeletal muscle index (SMI) in pre-frail or frail elderly individuals to investigate the impact of continuous HSP intake on ASM.

## 2. Materials and Methods

### 2.1. Study Design and Intervention

This study was a 12-week randomized, double-blind, parallel-group controlled trial, and details of the study design have been published [8]. Community-dwelling elderly subjects were recruited as paid volunteers between 2 May 2022 and 27 June 2022 by Macromill, Inc. (Tokyo, Japan), ASMARQ Co., Ltd. (Tokyo, Japan), and Nambu Co., Ltd. (Aichi, Japan). Both pre-frail and frail individuals were eligible, as frailty is recognized as a dynamic continuum in which pre-frailty represents a clinically meaningful stage for early intervention, so including both groups allowed us to capture the broader spectrum of vulnerability. Recruitment criteria were designed to reflect the real-world distribution of frailty in the community, where pre-frailty is more prevalent than frailty; consequently, the majority of participants in our sample were pre-frail. A total of 105 adults aged 65 to 83 years were screened for at least one of the criteria from the revised Japanese version of the Cardiovascular Health Study (hereinafter referred to as the J-CHS criteria) [9]. Of the five J-CHS evaluation criteria (walking speed, grip strength, physical activity, fatigue, and weight loss), those who met three or more criteria were defined as frail, and those who met one or two were defined as pre-frail. We allocated study subjects to an HSP soymilk group (HSP group: *n* = 36) with an intake of 14.5 g/200 mL soy protein (160 kcal of energy, 14.5 g of protein, 9.3 g of fat, 4.9 g of carbohydrates, 0.8 g of fiber; MARUSAN-AI Co., Ltd., Okazaki, Japan) and a control group (C group: *n* = 37) with an intake of 3.2 g/200 mL soy protein (137 kcal of energy, 3.2 g of protein, 2.1 g of fat, 26.7 g of carbohydrates, and 0.6 g of fiber). Random allocation to the two groups was performed using software (STATA; Version 14, StataCorp, College Station, TX, USA).

### 2.2. Measurement

Appendicular skeletal muscle mass (ASM) and the skeletal muscle index (SMI) were assessed using bioelectrical impedance analysis (BIA) conducted with the InBody570 instrument (InBody Co., Ltd., Seoul, Republic of Korea), a multifrequency segmental BIA system. ASM (kg) is the sum of upper and lower limb lean mass, and the SMI (kg/m^2^) is calculated by dividing ASM by height squared. Steps and total energy expenditure (TEE) were monitored using the Calorhythm Slim AM-122 (TANITA Corporation, Tokyo, Japan), and estimated protein intake (EPI) was calculated using the Maloni formula [10], which assumes that urinary nitrogen accounts for the majority of nitrogen loss and calculates daily protein intake from 24 h urinary urea nitrogen (UUN) and creatinine values. The UUN was calculated using the Tanaka formula [11], which calculates daily UUN excretion from spot urine samples based on the urinary urea and creatinine concentrations and body weight. Total energy intake (TEI) was determined by using a simplified self-administered dietary history questionnaire [12].

### 2.3. Outcomes

The outcome of this study was the change in the SMI from baseline at week 12. The primary (walking speed) and secondary (skeletal muscle mass) endpoints were prespecified in the study protocol, but subgroup analyses of the SMI and frailty status were performed post hoc in this study.

### 2.4. Subgroup Cutoff Values

A decrease in walking speed is one of the J-CHS criteria, with a cutoff value set to 1.0 m/s [9].

Step count is a popular indicator for assessing physical activity. Based on Tudor-Locke et al.’s classification of daily physical activity levels based on step count, the cutoff value was set to 5000 steps/day, with a count below this considered a “sedentary lifestyle”, in which the daily physical activity level is low with little engagement in exercise or sports, while a step count of the cutoff value or more was considered low-to-high-level exercise or sports [13].

Physical activity is a J-CHS criteria, and TEE is an indicator of physical activity. Based on age-specific TEE data, the cutoff value was set to 30 kcal/kg body weight/day ([14], p. 79).

TEI is strongly associated with frailty onset and progression in elderly people, and sufficient intake may have a preventative effect. In particular, low energy intake has been reported to be associated with an increased risk of frailty [15]. Based on data on elderly Japanese people’s estimated energy requirements, the cutoff value was set to 30 kcal/kg body weight/day [16].

Adequate protein intake has been reported to contribute to maintaining muscle mass and preventing frailty [17,18]. To prevent the onset of frailty and sarcopenia, it is recommended that Japanese elderly individuals (aged 65 years or older) consume at least 1.0 g/kg body weight/day of protein ([14], p. 114); therefore, the cutoff value for estimated protein intake (EPI) was set to 1.0 g/kg body weight/day.

### 2.5. Statistical Analysis

In this study, data on the SMI were missing for two cases, resulting in a total of 68 subjects being included in the analysis. A full analysis of all subjects (*n* = 68, HSP group: *n* = 34, control group: *n* = 34) and a subgroup analysis focusing on the 85% classified as pre-frail elderly individuals (*n* = 58, HSP group: *n* = 29, control group: *n* = 29) were performed. The following nine subgroup analyses were conducted: frailty phenotype, basal metabolic rate, walking speed, step count, TEE, TEI, estimated protein intake, age, and sex. An analysis of covariance was conducted to evaluate differences in SMI change, adjusting for age, sex, and baseline values. When regression line parallelism was not satisfied, Student’s *t*-test or the Mann–Whitney U test were performed; furthermore, these two tests were used to analyze the continuous variables of baseline data for pre-frail elderly subjects, while Fisher’s exact test was used for categorical variables and multiple regression analysis was performed for interactions. All subgroup analyses were exploratory in nature and were not adjusted for multiple comparisons. Two-tailed tests with a significance level of less than 5% were evaluated as “statistically significant differences”, and all statistical analyses were performed using EZR (Saitama Medical Center, Jichi Medical University, Saitama, Japan, Version 1.68), a graphical user interface for R (The R Foundation for Statistical Computing, Vienna, Austria, version 4.3.1) [19].

## 3. Results

### 3.1. Analysis of All Subjects

Of the 70 subjects analyzed, 2 had missing SMI data, resulting in 68 valid subjects for analysis (HSP group: *n* = 34, C group: *n* = 34), of which 58 were pre-frail and 10 were frail elderly subjects. No significant differences were observed in the baseline values of background data between the HSP group (*n* = 34) and C group (*n* = 34) (Table 1).

Subsequently, in the overall analysis (primary combined cohort), there was no significant change in the SMI between the HSP and C groups (*p* = 0.18); however, in the subsequent frailty phenotype (pre-frail and frail) subgroup analysis, interestingly, we observed a significant increase in the SMI in the HSP group compared to the C group (mean difference in SMI change: 0.08 kg/m^2^ [95%CI 0.004, 0.15], *p* = 0.044) (Figure 1A). Furthermore, in the following three subgroup analyses, the HSP group showed a significant increase in the SMI compared with the C group: walking speed ≥ 1.0 m/s (0.11 kg/m^2^ [95%CI 0.01, 0.21], *p* = 0.031), step count ≥5000 steps/day (0.15 kg/m^2^ [95%CI 0.03, 0.26], *p* = 0.016), and TEI ≥ 30 kcal/kg BW/day (0.14 kg/m^2^ [95%CI 0.02, 0.25], *p* = 0.019).

### 3.2. Analysis of Pre-Frail Subjects

Therefore, we also conducted a subgroup analysis limited to the pre-frail elderly individuals (*n* = 58), who accounted for 85% of the total subjects (*n* = 68). There were no significant differences in baseline data between the HSP (*n* = 29) and C (*n* = 29) groups (Table 2).

As shown in Figure 1B, the HSP group demonstrated a significant increase in the SMI compared to the C group in the following four subgroup analyses: ≥5000 steps/day (0.16 kg/m^2^ [95%CI 0.04, 0.27], *p* = 0.010), TEE ≥ 30 kcal/kg BW/day (0.14 kg/m^2^ [95%CI 0.004, 0.27], *p* = 0.045), TEI ≥ 30 kcal/kg BW/day (0.14 kg/m^2^ [95%CI 0.02, 0.26], *p* = 0.021), and male sex (0.12 kg/m^2^ [95%CI 0.01, 0.22], *p* = 0.029).

## 4. Discussion

In this study, we measured the change in the SMI in Japanese pre-frail or frail elderly individuals with ordinary lives to evaluate the effect of routine HSP soymilk intake on ASM.

As a result, while no significant differences were observed between the groups in the overall analysis of pre-frail and frail elderly subjects (*n* = 68), the subgroup analysis showed that among the pre-frail individuals, the HSP group had a significantly increased change in the SMI compared to the C group. Furthermore, in the three subgroups, walking speed ≥ 1.0 m/s, step count ≥ 5000 steps/day, and TEI ≥ 30 kcal/kg body weight/day, the HSP group also showed a significantly increased change in the SMI.

Of the subjects analyzed (*n* = 68), 85% were pre-frail elderly individuals (*n* = 58), so a subgroup analysis was performed to narrow the analysis to these subjects alone. The results showed that among the subgroups with steps ≥ 5000/day, TEE ≥ 30 kcal/kg body weight/day, TEI ≥ 30 kcal/kg body weight/day, and male sex, the HSP group showed a significantly greater change in the SMI compared to the C group; therefore, habitual intake of HSP soymilk may be related to ASM in pre-frail elderly individuals, though further confirmation is needed.

The association between soy protein intake and increased muscle mass (SMI/ASM) observed in this study is consistent with existing findings. Soy protein is rich in essential amino acids, particularly leucine, and has been reported to promote muscle protein synthesis via the mTOR signaling pathway [20].

In addition, animal studies have reported the effects of soy protein, including the suppression of muscle atrophy from disuse via the ubiquitin ligase system [21] and muscle atrophy in ovariectomized (OVX) mice, through soymilk consumption [22].

Furthermore, sufficient plant-based protein intake has been shown to contribute to maintaining and increasing muscle mass in elderly individuals [23]. These mechanistic backgrounds support the biological plausibility of the increased SMI observed in this study.

A previous study conducted by our team has also shown that high soy protein intake contributes to improved walking speed [8], which is a functional indicator reflecting neuromuscular function and muscle strength [24,25]; the SMI/ASM is a structural indicator of muscle mass, so the SMI increase observed in this study may therefore represent a structural improvement. That is, soy protein intake may support functional performance by increasing muscle mass, suggesting it could contribute to muscle health in the elderly through both structural and functional indicators.

The results of a recent meta-analysis of randomized controlled trials suggested that slightly increasing protein intake for several months, even by as little as 0.1 g/kg BW/d, may increase or help maintain muscle mass with or without resistance training [26]. Furthermore, in another meta-analysis of subjects over 60 years old, a soy protein (0.6–60 g/day) intervention was reported to improve lean muscle mass, and its benefits were equivalent to those of other interventions, such as animal protein consumption, exercise only, and exercise plus animal protein consumption [27].

Therefore, in this study, HSP soymilk intake (soy protein content: 14.5 g/200 mL) was 200 mL per day, and the soy protein intake load was 14.5 g/day (approximately 0.25 g/kg body weight/day), which was presumably sufficient to improve ASM.

Indeed, consistent with these meta-analyses, in the pre-frail elderly individual subgroup representing 58 out of 68 subjects in our study, adequate HSP soymilk ingestion for at least 12 weeks was associated with improvement in ASM, even without a resistance exercise training intervention.

Furthermore, compared to the C group, the subgroup of the HSP group with a significant increase in the SMI was analyzed, suggesting that the effects of continuous HSP intake were more likely to be noticeable on ASM in the elderly individuals. Step counts ≥ 5000 steps/day and TEE ≥ 30 kcal/kg BW/day were inferred to be subgroups in which physical activity in one’s daily life was maintained. In addition, it was also deduced that TEI ≥ 30 kcal/kg BW/day was a subgroup in which nutritional intake met the estimated daily energy requirement; however, due to the small sample size of women (*n* = 9), the reliability of the statistical analysis results was low, and further investigation is required.

We also noticed a significant increase in SMI change in males in the HSP group, though there was only a small proportion of female subjects in our subgroup analysis. This difference in the SMI in males should be validated in a larger intervention study to consider whether this male advantage is a sex-based difference. This HSP soymilk intervention demonstrated results consistent with the NILS-LSA study (National Institute of Geriatrics and Gerontology Longitudinal Study on Aging), a large-scale cohort study of Japanese older adults. Specifically, the NILS-LSA study indicated that TEE, energy intake, and protein intake were risk factors for low muscle mass in both men and women [28]; furthermore, while the SMI was a risk factor associated with frailty in men, no such association was observed in women [29].

In the primary combined cohort, soy protein intake did not improve the SMI. One possible explanation for the preferential benefit observed in pre-frail but not frail individuals is that the former may retain greater physiological reserves and anabolic responsiveness, whereas the latter may have a diminished capacity to respond to nutritional interventions.

In addition, daily physical activity is known to stimulate muscle protein synthesis and enhance amino acid utilization, suggesting a potential synergistic interaction with protein intake. These considerations remain speculative and highlight the need for further confirmatory studies [30,31].

In the primary analysis of the combined cohort, continuous HSP soymilk intake did not result in a significant improvement in SMI, indicating a lack of effect in the overall population and that the intervention did not demonstrate efficacy in the prespecified secondary endpoint.

However, exploratory subgroup analyses revealed significant increases in the SMI in specific subgroups, such as pre-frail individuals and those with higher physical activity or energy intake. These results suggest that the intervention may have beneficial effects under certain conditions, although such findings should be interpreted with caution given their post hoc nature.

### Limitations

This study has several limitations that should be acknowledged: first, subgroup analyses of the SMI and frailty status were conducted post hoc; therefore, they are exploratory in nature and need to be confirmed in future studies using prespecified endpoints. Second, the statistical power was constrained by the relatively small sample size, and the extensive subgroup analyses were performed without adjustment for multiple comparisons, raising the possibility of type I error. Third, body composition was assessed using bioelectrical impedance analysis (BIA) rather than dual-energy X-ray absorptiometry (DXA), which is considered the gold standard, so this may have introduced measurement bias. Fourth, the intervention period was limited to 12 weeks, which restricts conclusions regarding longer-term effects. Finally, the study population was relatively small and homogeneous, thereby limiting the generalizability of the findings to broader and more diverse populations.

## 5. Conclusions

In this study, no significant improvement in the SMI was observed in all subjects (frail and pre-frail elderly individuals); however, an exploratory subgroup analysis suggested that continued HSP soymilk intake in daily life was associated with potential benefits in pre-frail elderly individuals with high physical activity and energy intake levels. This study was conducted as an exploratory investigation, and its findings should be interpreted with caution. Further large-scale and systematic research is needed to confirm the reproducibility of the results.

## Figures and Tables

**Figure 1 nutrients-17-03900-f001:**
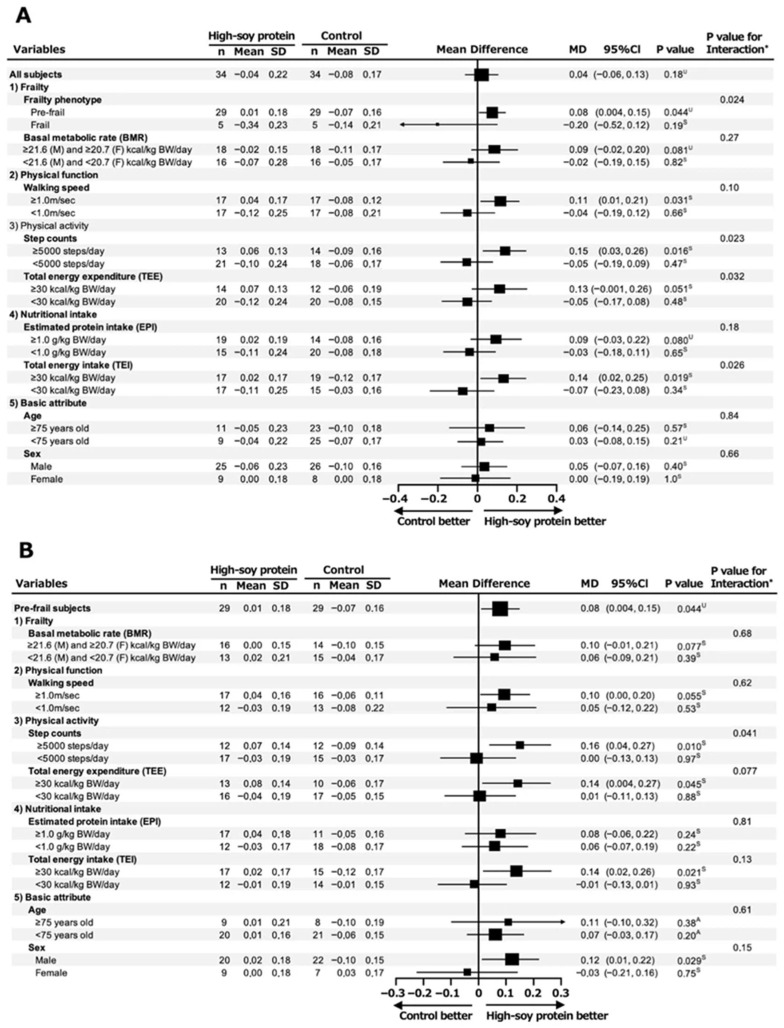
Forest plot of subgroup analysis for change in SMI at week 12 of intervention in pre-frail and frail elderly subjects. Subgroup analysis of SMI change in all subjects (pre-frail and frail elderly individuals) (**A**) and pre-frail subjects (**B**) at week 12 of intervention. Both subgroup analyses were performed using subjects’ baseline characteristics. ^A^ analysis of covariance; ^S^ Student’s *t*-test; ^U^ Mann–Whitney U test; * Multiple regression analysis.

**Table 1 nutrients-17-03900-t001:** Baseline characteristics of pre-frail and frail elderly participants for SMI analysis.

	High-Soy-Protein Group		Control Group		*p* Values	
	*n* = 34		*n* = 34			
Age (years)	72.0	(4.9)	71.6	(5.1)	0.71	^U^
Sex	Male 25	Female 9	Male 26	Female 8	1.00	^F^
Weight (kg)	65.9	(14.4)	60.9	(10.3)	0.24	^U^
BMI (kg/m^2^)	24.2	(4.5)	23.3	(2.9)	0.67	^U^
SMI (kg/m^2^)	7.16	(1.2)	7.03	(0.8)	1.00	^U^
Walking speed (m/sec)	1.1	(0.2)	1.1	(0.2)	0.91	^U^
Grip strength (kg)	32.6	(8.2)	31.7	(7.8)	0.65	^S^
Applicable participants in the revised J-CHS criteria						
Pre-frail, *n* (%)	29 (85.3) [1: *n* = 20, 2: *n* = 9]		29 (85.3) [1: *n* = 17, 2: *n* = 12]			
Frail, *n* (%)	5 (14.7) [3: *n* = 4, 4: *n* = 1]		5 (14.7) [3: *n* = 5]			
Number of applicable items in the revised J-CHS criteria						
Walking speed, *n* (%)	17	(50.0)	17	(50.0)	1.00	^F^
Grip strength, *n* (%)	2	(5.9)	4	(11.8)	0.67	^F^
Physical activity, *n* (%)	16	(47.1)	18	(52.9)	0.81	^F^
Fatigue, *n* (%)	11	(32.4)	11	(32.4)	1.00	^F^
Weight loss, *n* (%)	8	(23.5)	6	(17.6)	0.77	^F^
Hasegawa’s dementia scale—revised (HDS-R)	28.9	(1.7)	28.8	(1.4)	0.44	^U^
Physical activity						
Total energy expenditure (kcal/kg BW/day)	28.4	(4.2)	29.4	(4.6)	0.91	^U^
Step count (steps/day)	4573	(3133)	5359	(4349)	0.76	^U^
Nutritional intake						
Total energy intake (kcal/kg BW/day)	30.6	(13.2)	35.0	(15.7)	0.21	^U^
Estimated protein intake (g/kg BW/day)	1.05	(0.3)	1.05	(0.3)	0.96	^U^
Medical history	25	(73.5)	28	(82.4)	0.56	^F^
Currently being treated for	15	(44.1)	15	(44.1)	1.00	^F^
Hypertension	6	(17.6)	8	(23.5)		
Diabetes	6	(17.6)	5	(14.7)		
Dyslipidemia	1	(2.9)	1	(2.9)		
Chronic heart failure	1	(2.9)	0	(0.0)		
Angina	1	(2.9)	0	(0.0)		
Hyperuricemia	1	(2.9)	1	(2.9)		
Osteoporosis	0	(0.0)	1	(2.9)		
Rheumatism	0	(0.0)	1	(2.9)		
Chronic thyroiditis	0	(0.0)	1	(2.9)		
Glaucoma	1	(2.9)	1	(2.9)		

Mean (SD); ^S^ Student’s *t*-test; ^U^ Mann–Whitney U test; ^F^ Fisher’s exact test.

**Table 2 nutrients-17-03900-t002:** Baseline characteristics of pre-frail elderly participants for SMI analysis.

	High-Soy-Protein Group		Control Group		*p* Values	
	*n* = 29		*n* = 29			
Age (years)	71.8	(5.0)	72.1	(5.1)	0.76	^U^
Sex	Male 20	Female 9	Male 22	Female 7	0.77	^F^
Weight (kg)	63.7	(11.4)	61.2	(11.0)	0.41	^S^
BMI (kg/m^2^)	23.7	(3.5)	23.4	(2.9)	0.73	^S^
SMI (kg/m^2^)	7.0	(1.0)	7.1	(0.9)	0.80	^U^
Walking speed (m/s)	1.1	(0.2)	1.1	(0.2)	0.95	^U^
Grip strength (kg)	32.4	(8.5)	32.4	(7.9)	0.97	^S^
Number of applicable items in the revised J-CHS criteria						
Walking speed, *n* (%)	12	(41.4)	13	(44.8)	1.00	^F^
Grip strength, *n* (%)	2	(6.9)	2	(6.9)	1.00	^F^
Physical activity, *n* (%)	11	(37.9)	14	(48.3)	0.60	^F^
Fatigue, *n* (%)	7	(24.1)	7	(24.1)	1.00	^F^
Weight loss, *n* (%)	6	(20.7)	5	(17.2)	1.00	^F^
Hasegawa’s dementia scale—revised (HDS-R)	28.9	(1.7)	28.7	(1.4)	0.19	^U^
Physical activity						
Total energy expenditure (kcal/kg BW/day)	29.1	(3.5)	29.7	(4.1)	0.59	^S^
Step count (steps/day)	4852	(3245)	5620	(4469)	0.70	^U^
Nutritional intake						
Total energy intake (kcal/kg BW/day)	32.4	(13.2)	33.9	(15.9)	0.77	^U^
Estimated protein intake (g/kg BW/day)	1.1	(0.3)	1.0	(0.3)	0.32	^U^
Medical history	21	(72.4)	26	(89.7)	0.18	^F^
Currently being treated for	13	(44.8)	16	(55.2)	1.00	^F^
Hypertension	4	(13.8)	10	(34.5)		
Diabetes	5	(17.2)	5	(17.2)		
Dyslipidemia	0	(0.0)	1	(3.4)		
Chronic heart failure	0	(0.0)	0	(0.0)		
Angina	1	(3.4)	0	(0.0)		
Hyperuricemia	1	(3.4)	1	(3.4)		
Osteoporosis	0	(0.0)	1	(3.4)		
Rheumatism	0	(0.0)	1	(3.4)		
Chronic thyroiditis	0	(0.0)	1	(3.4)		
Glaucoma	1	(3.4)	1	(3.4)		

Mean (SD); ^S^ Student’s *t*-test; ^U^ Mann–Whitney U test; ^F^ Fisher’s exact test.

## Data Availability

The datasets generated and analyzed in this study are available upon reasonable request to the corresponding author. The data are not publicly available due to the protection of subjects’ privacy.

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
