# Peer review of "The Effect of Routine High-Soy-Protein Soymilk Intake on the Skeletal Muscle Index (SMI) in Japanese Pre-Frail Elderly Individuals with an Ordinary Life: A Post Hoc Analysis of a Randomized Controlled Trial"

_nutrients, 2025, doi:10.3390/nu17243900_

Round 1
Reviewer 1 Report
Comments and Suggestions for Authors
This study was as a secondary analysis of authors’ previous study. They conducted further statistical analysis focusing on changes in skeletal muscle index (SMI) in pre-frail or frail elderly people to investigate the impact of continuous intake of high-soy protein (HSP) on changes in appendicular skeletal muscle mass (ASM), a secondary endpoint. The authors concluded that continuous intake of HSP soymilk in daily life may improve ASM and ameliorate physical frailty in the Japanese pre-frail elderly.
Comments
The reviewer has some concerns as follows:
- The authors claimed this study was a 12-week randomized, double-blind, parallel-group controlled trial, and details of the study design have been published (Sato et al., 2025; ref. 5 of this study). However, some questions need clarification:
(1) If it is the same clinical trial, why are the sample sizes different in the two studies (all subjects: this study: n=68, HSP group: n=34, control group: n=34; Sato et al., 2025: n=73, HSP group: n = 36, control group: n = 37)?
(2) How were patients selected?
(3) Are they under the same IRB approval number?
- How to distinguish between the pre-frail and frail elderly participants? What are the criteria used to define this distinction? It needs to be explained.
- Please correct the italic fonts in the Title and the paragraph in lines 79-86.
- In line 49, the full name for “ABC” needs to be described.
- There is only one self-citation in the reference list and it can be accepted. The references cited in this manuscript appropriate and relevant to this research.
- Overall, this manuscript needs a revision.
Author Response
The reviewer has some concerns as follows:
- The authors claimed this study was a 12-week randomized, double-blind, parallel-group controlled trial, and details of the study design have been published (Sato et al., 2025; ref. 5 of this study). However, some questions need clarification:
(1) If it is the same clinical trial, why are the sample sizes different in the two studies (all subjects: this study: n=68, HSP group: n=34, control group: n=34; Sato et al., 2025: n=73, HSP group: n = 36, control group: n = 37)?
Response:
We appreciate your valuable comments and suggestions to improve our manuscript. The reason for the different sample sizes of the two studies has been added to the Statistical Analysis section as follows (lines 155-156):
In this study, data on SMI was missing for two cases, resulting in a total of 68 subjects included in the analysis.
(2) How were patients selected?
Response:
Thank you very much for your careful comment. Subject recruitment has been added to the Methods section as follows (lines 89-91):
Subjects were recruited as paid volunteers between May 2, 2022 and June 27, 2022 by Macromill, Inc. (Tokyo), ASMARQ Co., Ltd. (Tokyo), and Nambu Co., Ltd. (Aichi).
(3) Are they under the same IRB approval number?
Response:
We are grateful for your careful reading. This study and Reference 5 (2025) are under the same IRB approval number (2142) from the Healthcare Systems Ethics Committee. The IRB approval number is listed in the Institutional Review Board Statement (lines 332-335).
- How to distinguish between the pre-frail and frail elderly participants? What are the criteria used to define this distinction? It needs to be explained.
Response:
Thank you very much for your careful reading and we really appreciate and agree with your comments. The definitions of the pre-frail and frail elderly have been added to the Methods section as follows (lines 99-102):
Of the five evaluation criteria from J-CHS criteria (walking speed, grip strength, physical activity, fatigue, and weight loss), those who met three or more criteria were defined as frail, and those who met one or two criteria were defined as pre-frail.
- Please correct the italic fonts in the Title and the paragraph in lines 79-86.
Response:
Thank you very much for your careful comment. After uploading the manuscript to the system, the font style was changed to italic. We inquired with Assistant Editor Ms. Saphira Wang about this error, and she replied, "We applied a temporary layout when we received your manuscript, so there may be some formatting errors."
- In line 49, the full name for “ABC” needs to be described.
Response:
Thank you very much for your careful comment. In line 52, we have added the full name of ABC: the Health, Aging, and Body Composition.
- There is only one self-citation in the reference list and it can be accepted. The references cited in this manuscript appropriate and relevant to this research.
Response:
We really appreciate your comments.
- Overall, this manuscript needs a revision.
Response:
Thank you very much for your careful reading. We deeply appreciate and agree with your comments. We have clearly stated the limitations of this study and revised the title, abstract, results, discussion, and conclusion sections as follows:
Revised Limitations:
The limitations section has been added as follows (lines 303-314):
This study has several limitations that should be acknowledged. First, subgroup analyses of SMI and frailty status were conducted post hoc. Therefore, these results are exploratory in nature and need to be confirmed in future studies using prespecified endpoints. Second, the statistical power was constrained by the relatively small sample size, and the extensive subgroup analyses were performed without adjustment for multiple comparisons, raising the possibility of type I error. Third, body composition was assessed using bioelectrical impedance analysis (BIA) rather than dual-energy X-ray absorptiometry (DXA), which is considered the gold standard and may have introduced measurement bias. Fourth, the intervention period was limited to 12 weeks, which restricts conclusions regarding longer-term effects. Finally, the study population was relatively small and homogeneous, thereby limiting the generalizability of the findings to broader and more diverse populations.
Rationale:
A detailed limitations section was added to acknowledge methodological constraints, including the exploratory nature of subgroup analyses, limited sample size, and use of BIA instead of DXA. This addition was made to provide a balanced interpretation and to strengthen the scientific rigor of the manuscript.
Revised Title (lines 3-5):
The effect of routine high-soy protein soymilk intake on skeletal muscle index (SMI) in the Japanese pre-frail elderly with ordinary life: a post hoc analysis of a randomized controlled trial
Rationale:
The title was revised to better convey the study population, the intervention, and the post hoc nature of the analysis. This clarification was intended to make the scope and design of the study more transparent for readers.
Revised Abstract:
The Abstract has been revised as follows and:
Results (lines 30-31): In the overall analysis (primary combined cohort), SMI showed no significant change between the HSP soymilk and control soymilk groups (P=0.18). However,
Conclusion (lines 37-41): In this study, continuous intake of HSP soy milk did not improve SMI in the primary combined cohort. However, exploratory subgroup analyses suggested potential benefits in pre-frail elderly and elderly individuals with high levels of physical activity and energy intake. These findings are preliminary and require further research.
Rationale:
The abstract was modified to clearly indicate that the primary analysis showed no significant effect, while exploratory subgroup analyses suggested potential benefits under specific conditions. This change was made to ensure an accurate summary and avoid overstating the findings.
Revised Results:
The following sentence was added to the Results section (line 184):
In the overall analysis of all subjects (primary combined cohort),
Rationale:
An explicit statement was added to the Results section to clarify that the primary combined cohort analysis did not show a significant improvement in SMI. This was done to maintain consistency between the abstract and main text and to improve transparency.
Revised Discussion:
The following sentence was revised to the Discussion section (lines 234-235):
Habitual intake of HSP soymilk may be related to ASM in pre-frail elderly, though further confirmation is needed.
The following sentence was added to the Discussion section (lines 292-300):
In the primary analysis of the combined cohort, continuous intake of HSP soymilk did not result in a significant improvement in SMI, indicating a lack of effect in the overall population. This finding indicates that the intervention did not demonstrate efficacy in the prespecified secondary endpoint.
However, exploratory subgroup analyses revealed significant increases in SMI in specific subgroups, such as pre-frail individuals and those with higher physical activity or energy intake. These results suggest that the intervention may have beneficial effects under certain conditions, although such findings should be interpreted with caution given their post hoc nature.
Rationale:
The Discussion was revised and expanded to emphasize that the intervention did not demonstrate efficacy in the overall population, and that subgroup findings should be interpreted with caution. This was intended to provide a balanced perspective and avoid overgeneralization.
Revised Conclusion:
The Conclusion section has been revised as follows (lines 317-320):
In this study, no significant improvement in SMI was observed in all subjects (frail and pre-frail elderly). However, exploratory subgroup analysis suggested that continued intake of HSP soymilk in daily life was associated with potential benefits in pre-frail elderly with high levels of physical activity and energy intake.
Rationale:
The conclusion was rewritten to clearly state that no significant improvement was observed in the overall population, while potential benefits were suggested in specific subgroups. This change was made to ensure that the conclusion accurately reflects the study findings.

Reviewer 2 Report
Comments and Suggestions for Authors
The paper is clear, well-structured but there are several stylistic, methodological, and editorial issues that should be corrected before publication:
-please clearly that all subgroup analyses are exploratory
-adjust for multiple testing, or downgrade the strength of conclusions
- include a section explicitly outlining the prespecified vs. post-hoc nature of analyses
-because of bias concerns please describe clearly independent from the sponsor
-strengthen the description of blinding procedures
- explain biological or clinical rationale for each cutoff
-please emphasize clearly Lack of Effect in the Primary Combined Cohort
-to increase value and interest for reader consider to address why soy protein might preferentially benefit pre-frail but not frail individuals and how daily activity interacts biologically with protein intake
Comments on the Quality of English LanguageEnglish language editing is required
Author Response
The paper is clear, well-structured but there are several stylistic, methodological, and editorial issues that should be corrected before publication:
- - please clearly that all subgroup analyses are exploratory
Response:
We appreciate your valuable comments and suggestions to improve our manuscript. We have added the following sentence to the Statistical Analysis section (lines 165-166):
All subgroup analyses were exploratory in nature and were not adjusted for multiple comparisons.
- - adjust for multiple testing, or downgrade the strength of conclusions
Response:
We appreciate your valuable comments and suggestions to improve our manuscript. The Conclusions section of this study have been revised as follows (lines 317-320):
In this study, no significant improvement in SMI was observed in all subjects (frail and pre-frail elderly). However, exploratory subgroup analysis suggested that continued intake of HSP soymilk in daily life was associated with potential benefits in pre-frail elderly with high levels of physical activity and energy intake.
The conclusion of the Abstract has also been revised as follows (lines 37-41):
In this study, no significant improvement in SMI was observed in all subjects (frail and pre-frail elderly). However, exploratory subgroup analysis suggested that continued intake of HSP soymilk in daily life was associated with potential benefits in pre-frail elderly with high levels of physical activity and energy intake. These findings are preliminary and require further research.
- - include a section explicitly outlining the prespecified vs. post-hoc nature of analyses
Response:
We appreciate your valuable comments and suggestions to improve our manuscript. To clearly distinguish between pre-specified and post hoc analyses, we have revised the Materials and Methods, Discussion, and Limitations sections as follows.
Materials and Methods (lines 126-129):
The primary endpoint (walking speed) and secondary endpoint (skeletal muscle mass) were pre-specified in the study protocol, but subgroup analyses of SMI and frailty status were performed post hoc in this study.
Limitations (lines 303-305):
This study has several limitations that should be acknowledged. First, subgroup analyses of SMI and frailty status were conducted post hoc. Therefore, these results are exploratory in nature and need to be confirmed in future studies using prespecified endpoints.
- - because of bias concerns please describe clearly independent from the sponsor
Response:
Thank you very much for your careful comment. Specifically, we have added the following sentence to COI (lines 344-346):
The sponsor had no role in the design of the study; in the collection, analysis, or interpretation of data; in the writing of the manuscript; or in the decision to submit the article for publication.
- -strengthen the description of blinding procedures
Response:
Thank you very much for your careful comment. We have added the following sentence to Materials and Methods section (lines 107-108):
Random allocation to the two groups was performed using software (STATA; Version 14, StataCorp, College Station, TX, USA).
- - explain biological or clinical rationale for each cutoff
Response:
We appreciate your valuable comments and suggestions to improve our manuscript. We have added the following explanation to the new “2.4. Each cutoff values for subgroups” section regarding the clinical rationale for each cutoff value (lines 131-152). Additionally, references (Ref. No. 15, 17, 18, 19) have been added to the References section.
2.4. Each cutoff values for subgroups
A decrease in walking speed is one of the criteria for assessing frail (J-CHS criteria), and the cutoff value for walking speed is 1.0 m/s [9].
Step count is one of the popular indicators for assessing physical activity. Based on Tudor-Locke et al.'s classification of daily physical activity levels based on step count, the cutoff value for step count was set at 5,000 steps/day. A step count of less than 5,000 steps/day is considered a "sedentary lifestyle" in which the daily physical activity level is low or high, with little exercise or sports, while a step count of 5,000 steps/day or more is considered a low- to high-level exercise or sports [13].
Physical activity is one of the frailty assessment criteria (J-CHS criteria), and total energy expenditure (TEE) is an indicator of physical activity. Based on age-specific TEE data, the cutoff value was set at 30 kcal/kg body weight/day [14].
Total energy intake (TEI) is strongly associated with the onset and progression of frailty in elderly people, and sufficient intake may have a preventative effect. In particular, low energy intake has been reported to be associated with an increased risk of frailty [15]. Based on data on the estimated energy requirements of elderly Japanese people, the cutoff value was set at 30 kcal/kg body weight/day [16].
Adequate protein intake has been reported to contribute to maintaining muscle mass and preventing frail [17, 18]. To prevent the onset of frail and sarcopenia, it is recommended that Japanese elderly (aged 65 years or older) consume at least 1.0 g/kg body weight/day of protein [19]. Therefore, the cutoff value for estimated protein intake (EPI) was set at 1.0 g/kg body weight/day.
Additional references (lines 389-390 and 393-399):
15. Lorenzo-López, L.; Maseda, A.; de Labra, C.; Regueiro-Folgueira, L.; Rodríguez-Villamil, J. L.; Millán-Calenti, J. C. Nutritional determinants of frailty in older adults: A systematic review. BMC Geriatr. 2017, 108.
17. Nanri, H.; Watanabe, D.; Yoshida, T.; Yoshimura, E.; Okabe, Y.; Ono, M.; Koizumi, T.; Kobayashi, H.; Fujita, H.; Kimura, M.; Yamada, Y. Adequate protein intake on comprehensive frailty in older adults: Kyoto-Kameoka study. J. Nutr. Health Aging 2022, 26, 161–168.
18. Mendonça, N.; Kingston, A.; Granic, A.; Jagger, C. Protein intake and transitions between frailty states and to death in very old adults: the Newcastle 85+ study. Age Ageing 2019, 49, 32–38.
19. Ministry of Health, Labour and Welfare. Report of the Study Group on the Formulation of Dietary Intakes for Japanese (2020 Edition) [in Japanese]. 2019, p 114. https://www.mhlw.go.jp/content/10904750/000586553.pdf
- -please emphasize clearly Lack of Effect in the Primary Combined Cohort
Response:
We appreciate your valuable comments and suggestions to improve our manuscript. We have revised the Abstract, Results, Discussion, and Conclusion sections to explicitly state that no significant differences were observed in the primary analysis of all subjects.
Revised Abstract:
The Abstract has been revised as follows (lines 30-31 and 37-41):
Results: In the overall analysis (primary combined cohort), SMI showed no significant change between the HSP soymilk and control soymilk groups (P=0.18). However,
Conclusion: In this study, continuous intake of HSP soy milk did not improve SMI in the primary combined cohort. However, exploratory subgroup analyses suggested potential benefits in pre-frail elderly and elderly individuals with high levels of physical activity and energy intake. These findings are preliminary and require further research.
Revised Results:
The following sentence was added to the Results section (line 184):
In the overall analysis of all subjects (primary combined cohort),
Revised Discussion:
The following sentence was revised to the Discussion section (lines 234-235):
Habitual intake of HSP soymilk may be related to ASM in pre-frail elderly, though further confirmation is needed.
The following sentence was added to the Discussion section (lines 292-300):
In the primary analysis of the combined cohort, continuous intake of HSP soymilk did not result in a significant improvement in SMI, indicating a lack of effect in the overall population. This finding indicates that the intervention did not demonstrate efficacy in the prespecified secondary endpoint.
However, exploratory subgroup analyses revealed significant increases in SMI in specific subgroups, such as pre-frail individuals and those with higher physical activity or energy intake. These results suggest that the intervention may have beneficial effects under certain conditions, although such findings should be interpreted with caution given their post hoc nature.
Revised Conclusion:
The Conclusion section has been revised as follows (lines 317-320):
In this study, no significant improvement in SMI was observed in all subjects (frail and pre-frail elderly). Exploratory subgroup analysis suggested that continued intake of HSP soymilk in daily life was associated with favorable changes in SMI and may contribute to improving physical frailty in the Japanese pre-frail elderly.
- -to increase value and interest for reader consider to address why soy protein might preferentially benefit pre-frail but not frail individuals and how daily activity interacts biologically with protein intake
Response:
Thank you for your insightful comments. Although the mechanism of the beneficial effects of soy protein in pre-frail individuals is unknown, we have described possible mechanisms in the Discussion section. Additionally, although direct evidence is limited, we have added a statement regarding the biological interaction between daily activity and protein intake to the Discussion section as follows (lines 283-291):
In the primary combined cohort, soy protein intake did not improve SMI. One possible explanation for the preferential benefit observed in pre-frail but not frail individuals is that pre-frail subjects may retain greater physiological reserve and anabolic responsiveness, whereas frail subjects may have diminished capacity to respond to nutritional interventions. In addition, daily physical activity is known to stimulate muscle protein synthesis and enhance amino acid utilization, suggesting a potential synergistic interaction with protein intake. These considerations remain speculative and highlight the need for further confirmatory studies [31, 32].
Additional references (lines 425-428):
31. Dideriksen, K.; Reitelseder, S.; Holm, L. Influence of amino acids, dietary protein, and physical activity on muscle mass development in humans. Nutrients 2013, 5, 852–876.
32. Ato, S.; Fujita, S. Regulation of muscle protein metabolism by nutrition and exercise. J. Phys. Fitness Sports Med. 2017, 6, 119–124.

Reviewer 3 Report
Comments and Suggestions for Authors
The topic is relevant to the field of nutrition and ageing, and the exploration of muscle mass outcomes in pre-frail and frail older adults is potentially valuable. However, several methodological and reporting issues require clarification.
Major points
The manuscript needs clearer transparency regarding its connection to the previously published RCT. It should state: a) Which outcomes were prespecified and which are post hoc. b) Which data have been previously published and which are new c) Why SMI/ASM were not reported in the original trial.
It is stated that the endpoint is ASM but it is simultaneously defined as the change in SMI, which is a different variable. ASM (kg) and SMI (kg/m²) are not interchangeable, and this inconsistency introduces methodological ambiguity.
Both pre-frail and frail participants were included, yet the sample is overwhelmingly pre-frail (85%), and most analyses and significant findings derive almost exclusively from this subgroup. This raises questions about the validity of combining pre-frail and frail individuals in the initial analyses, as well as about the external interpretability of the results. A clearer justification of the inclusion criteria and their practical application is needed.
Although the use of InBody 570 implies BIA, the manuscript does not explicitly state the technique used to assess SMI/ASM. This should be clearly reported, and the use of BIA instead of DXA acknowledged as a limitation, particularly in pre-frail/frail older adults.
Further methodological clarity is needed regarding: a) Units, formulas, and definitions used for SMI/ASM., b) Any reference cut-offs (EWGSOP/AWGS).
The manuscript relies heavily on subgroup findings, many of them small and not prespecified. No adjustment for multiple testing is applied, increasing the likelihood of chance findings. This substantially limits interpretability and should be acknowledged more explicitly.
The rationale linking soy protein intake to changes in SMI/ASM needs strengthening. The mechanistic context is limited, and the contrast between the original walking speed findings and the current muscle mass outcomes is not addressed.
The limitations section is insufficient and should be expanded to acknowledge the post hoc design and limited power, the extensive unadjusted subgroup analyses, the use of BIA instead of DXA, the short 12-week duration, and the small, homogeneous sample.
Minor points
Commercial devices and materials should be cited with full manufacturer information (company, city, country) at first mention.
The formulas used to calculate UUN (Tanaka) and protein intake (Maloni) are cited only by name. For transparency, the authors could consider briefly describing these formulas.
The subgroup thresholds used in the analyses (e.g., ≥5000 steps/day, ≥30 kcal/kg/day, ≥1.0 m/s walking speed) are reasonable, but the manuscript should briefly justify or cite the evidence supporting these cut-offs .
Adding a few recent studies on soy protein and muscle health would strengthen the background.
Author Response
The topic is relevant to the field of nutrition and ageing, and the exploration of muscle mass outcomes in pre-frail and frail older adults is potentially valuable. However, several methodological and reporting issues require clarification.
Major points
- The manuscript needs clearer transparency regarding its connection to the previously published RCT. It should state:
a) Which outcomes were prespecified and which are post hoc.
b) Which data have been previously published and which are new
c) Why SMI/ASM were not reported in the original trial.
a) Which outcomes were prespecified and which are post hoc.
Response:
We appreciate your valuable comments and suggestions to improve our manuscript. To clearly distinguish between pre-specified and post hoc analyses, we have revised the Materials and Methods section as follows (lines 126-129):
The outcome of this study was the change in SMI from baseline at week 12. The primary endpoint (walking speed) and secondary endpoint (skeletal muscle mass) were pre-specified in the study protocol, but subgroup analyses of SMI and frailty status were performed post hoc in this study.
b) Which data have been previously published and which are new
Response:
We appreciate your valuable comments. Walking speed of the primary endpoint and skeletal muscle mass of the secondary endpoints were reported in the previous paper [8].
The results of the SMI analysis are new data.
c) Why SMI/ASM were not reported in the original trial.
Response:
We appreciate your valuable comments. The reason why the analysis of SMI was not reported in the previous study [8] is because it was a post hoc analysis.
- It is stated that the endpoint is ASM but it is simultaneously defined as the change in SMI, which is a different variable. ASM (kg) and SMI (kg/m²) are not interchangeable, and this inconsistency introduces methodological ambiguity.
Response:
We appreciate your comment on this important point and agree with you. To clarify that SMI is a post-hoc analysis and to eliminate methodological ambiguity, we have clearly stated the limitations of this study and revised the title, abstract, results, discussion, and conclusion sections as follows:
Revised Title (lines 3-5):
The effect of routine high-soy protein soymilk intake on skeletal muscle index (SMI) in the Japanese pre-frail elderly with ordinary life: a post hoc analysis of a randomized controlled trial
Rationale:
The title was revised to better convey the study population, the intervention, and the post hoc nature of the analysis. This clarification was intended to make the scope and design of the study more transparent for readers.
Revised Abstract:
The Abstract has been revised as follows and:
Results (lines 30-31): In the overall analysis (primary combined cohort), SMI showed no significant change between the HSP soymilk and control soymilk groups (P=0.18). However,
Conclusion (lines 37-41): In this study, continuous intake of HSP soy milk did not improve SMI in the primary combined cohort. However, exploratory subgroup analyses suggested potential benefits in pre-frail elderly and elderly individuals with high levels of physical activity and energy intake. These findings are preliminary and require further research.
Rationale:
The abstract was modified to clearly indicate that the primary analysis showed no significant effect, while exploratory subgroup analyses suggested potential benefits under specific conditions. This change was made to ensure an accurate summary and avoid overstating the findings.
Revised Results:
The following sentence was added to the Results section (line 184):
In the overall analysis of all subjects (primary combined cohort),
Rationale:
An explicit statement was added to the Results section to clarify that the primary combined cohort analysis did not show a significant improvement in SMI. This was done to maintain consistency between the abstract and main text and to improve transparency.
Revised Discussion:
The following sentence was revised to the Discussion section (lines 234-235):
Habitual intake of HSP soymilk may be related to ASM in pre-frail elderly, though further confirmation is needed.
The following sentence was added to the Discussion section (lines 292-300):
In the primary analysis of the combined cohort, continuous intake of HSP soymilk did not result in a significant improvement in SMI, indicating a lack of effect in the overall population. This finding indicates that the intervention did not demonstrate efficacy in the prespecified secondary endpoint.
However, exploratory subgroup analyses revealed significant increases in SMI in specific subgroups, such as pre-frail individuals and those with higher physical activity or energy intake. These results suggest that the intervention may have beneficial effects under certain conditions, although such findings should be interpreted with caution given their post hoc nature.
Rationale:
The Discussion was revised and expanded to emphasize that the intervention did not demonstrate efficacy in the overall population, and that subgroup findings should be interpreted with caution. This was intended to provide a balanced perspective and avoid overgeneralization.
Revised Conclusion:
The Conclusion section has been revised as follows (lines 317-320):
In this study, no significant improvement in SMI was observed in all subjects (frail and pre-frail elderly). However, exploratory subgroup analysis suggested that continued intake of HSP soymilk in daily life was associated with potential benefits in pre-frail elderly with high levels of physical activity and energy intake.
Rationale:
The conclusion was rewritten to clearly state that no significant improvement was observed in the overall population, while potential benefits were suggested in specific subgroups. This change was made to ensure that the conclusion accurately reflects the study findings.
Revised Limitations:
The limitations section has been added as follows (lines 303-314):
This study has several limitations that should be acknowledged. First, subgroup analyses of SMI and frailty status were conducted post hoc. Therefore, these results are exploratory in nature and need to be confirmed in future studies using prespecified endpoints. Second, the statistical power was constrained by the relatively small sample size, and the extensive subgroup analyses were performed without adjustment for multiple comparisons, raising the possibility of type I error. Third, body composition was assessed using bioelectrical impedance analysis (BIA) rather than dual-energy X-ray absorptiometry (DXA), which is considered the gold standard and may have introduced measurement bias. Fourth, the intervention period was limited to 12 weeks, which restricts conclusions regarding longer-term effects. Finally, the study population was relatively small and homogeneous, thereby limiting the generalizability of the findings to broader and more diverse populations.
Rationale:
A detailed limitations section was added to acknowledge methodological constraints, including the exploratory nature of subgroup analyses, limited sample size, and use of BIA instead of DXA. This addition was made to provide a balanced interpretation and to strengthen the scientific rigor of the manuscript.
- Both pre-frail and frail participants were included, yet the sample is overwhelmingly pre-frail (85%), and most analyses and significant findings derive almost exclusively from this subgroup. This raises questions about the validity of combining pre-frail and frail individuals in the initial analyses, as well as about the external interpretability of the results. A clearer justification of the inclusion criteria and their practical application is needed.
Response:
We appreciate your thoughtful comments. We included both groups because frailty is a continuous spectrum, and pre-frailty is an important stage for early intervention. Because pre-frailty is more common in community-dwelling elderly, our results are primarily based on this group. We added in the Methods section as follows (lines 91-97):
Participants were recruited from community-dwelling elderly. Both pre-frail and frail individuals were eligible, as frailty is recognized as a dynamic continuum in which pre-frail represents a clinically meaningful stage for early intervention. Including both groups allowed us to capture the broader spectrum of vulnerability. Recruitment criteria were designed to reflect the real-world distribution of frailty in the community, where pre-frail is more prevalent than frail. Consequently, the majority of participants in our sample were pre-frail.
- Although the use of InBody 570 implies BIA, the manuscript does not explicitly state the technique used to assess SMI/ASM. This should be clearly reported, and the use of BIA instead of DXA acknowledged as a limitation, particularly in pre-frail/frail older adults.
Response:
Thank you for your insightful comment. We agree that the measurement technique for SMI/ASM should be clearly stated. Accordingly, we have revised the Methods section to explicitly state that SMI and ASM were assessed using bioelectrical impedance analysis (BIA) with the InBody 570 device.
In addition, we have added a sentence to the Limitations section acknowledging the use of BIA instead of dual-energy X-ray absorptiometry (DXA), particularly noting its potential limitations in pre-frail and frail older adults. The revised text reads:
Revised Methods (lines 111-113):
Appendicular skeletal muscle mass (ASM) and skeletal muscle index (SMI) were assessed using bioelectrical impedance analysis (BIA) with the InBody570 instrument (InBody Co., Ltd., Seoul, Korea), a multifrequency segmental BIA system.
Revised Limitations (lines 308-311):
Third, body composition was assessed using bioelectrical impedance analysis (BIA) rather than dual-energy X-ray absorptiometry (DXA), which is considered the gold standard and may have introduced measurement bias.
- Further methodological clarity is needed regarding: a) Units, formulas, and definitions used for SMI/ASM., b) Any reference cut-offs (EWGSOP/AWGS).
Response:
Thank you for pointing out the need for clarification of our methodology.
a) We have added the units, formulas, and definitions for Appendicular Skeletal Muscle Mass (ASM) and Skeletal Muscle Index (SMI) to the Methods section as follows (lines 113-115):
ASM (kg) is the sum of the lean mass of the upper and lower limbs. SMI (kg/m²) is calculated by dividing ASM by height squared.
b) Since this study evaluated changes in SMI, no cutoff values were used.
- The manuscript relies heavily on subgroup findings, many of them small and not prespecified. No adjustment for multiple testing is applied, increasing the likelihood of chance findings. This substantially limits interpretability and should be acknowledged more explicitly.
Response:
We appreciate your thoughtful comments regarding limitations. In response, we added to more clearly acknowledge the limitations of this study in Limitations section as follows (lines 306-308):
Second, the statistical power was constrained by the relatively small sample size, and the extensive subgroup analyses were performed without adjustment for multiple comparisons, raising the possibility of type I error.
- The rationale linking soy protein intake to changes in SMI/ASM needs strengthening. The mechanistic context is limited, and the contrast between the original walking speed findings and the current muscle mass outcomes is not addressed.
Response:
We really appreciate and agree with your comments. In response, we added the following to the discussion section regarding the evidence, mechanisms, and comparison with prior studies on improved walking speed concerning soy protein intake and changes in SMI/ASM (lines 236-252):
The association between soy protein intake and increased muscle mass (SMI/ASM) observed in this study is consistent with existing findings. Soy protein is rich in essential amino acids, particularly leucine, and has been reported to promote muscle protein synthesis via the mTOR signaling pathway [21].
In addition, animal studies have reported effects of soy protein, including suppression of disuse muscle atrophy via the ubiquitin ligase system [22] and suppression of muscle atrophy in ovariectomized (OVX) mice through soymilk consumption [23].
Furthermore, sufficient intake of plant-based protein has been shown to contribute to the maintenance and increase of muscle mass in elderly [24]. These mechanistic backgrounds support the biological plausibility of the increased SMI observed in this study.
Previous our study has also shown that high soy protein intake contributes to improved walking speed [8]. While walking speed is a functional indicator reflecting neuromuscular function and muscle strength [25, 26], SMI/ASM is a structural indicator of muscle mass. The SMI increase observed in this study may therefore represent a structural improvement. That is, soy protein intake may support functional performance by increasing muscle mass, suggesting it could contribute to muscle health in the elderly through both structural and functional indicators.
Additional references (lines 402-414):
21. Dijk, F. J.; van Dijk, M.; Roberts, J.; van Helvoort, A.; Furber, M. J. W. Pea and soy fortified with leucine stimulates muscle protein synthesis comparable to whey in a murine ageing model. Eur. J. Nutr. 2024, 64, 12.
22. Abe, T.; Kohno, S.; Yama, T.; Ochi, A.; Suto, T.; Hirasaka, K.; Ohno, A.; Teshima-Kondo, S.; Okumura, Y.; Oarada, M.; et al. Soy Glycinin Contains a Functional Inhibitory Sequence against Muscle-Atrophy-Associated Ubiquitin Ligase Cbl-b. Int. J. Endocrinol. 2013, Article ID 907565.
23. Kitajima, Y.; Ogawa, S.; Egusa, S.; Ono, Y. Soymilk Improves Muscle Weakness in Young Ovariectomized Female Mice. Nutrients 2017, 9, 834.
24. Olaniyan, E. T.; O’Halloran, F.; McCarthy, A. L. Dietary protein considerations for muscle protein synthesis and muscle mass preservation in older adults. Nutr. Res. Rev. 2021, 34, 147–157.
25. Zhang, Y.; Morita, M.; Hirano, T.; Doi, K.; Han, X.; Matsunaga, K.; Jiang, Z. A novel method for identifying frailty and quantifying muscle strength using the six-minute walking test. Sensors 2024, 24, 4489.
26. Cui, C.; Miao, H.; Liang, T.; Liu, X.; Liu, X. Analysis of muscle synergy and muscle functional network at different walking speeds based on surface electromyographic signal [in Chinese]. Sheng Wu Yi Xue Gong Cheng Xue Za Zhi 2023, 40, 938–944.
- The limitations section is insufficient and should be expanded to acknowledge the post hoc design and limited power, the extensive unadjusted subgroup analyses, the use of BIA instead of DXA, the short 12-week duration, and the small, homogeneous sample.
Response:
We appreciate your thoughtful comments regarding limitations. In response, we have created this new section to more clearly acknowledge the limitations of this study as follows (lines 303-314):
Limitations
This study has several limitations that should be acknowledged. First, the analyses were conducted in a post hoc manner, which may limit the strength of causal inferences. Second, the statistical power was constrained by the relatively small sample size, and the extensive subgroup analyses were performed without adjustment for multiple comparisons, raising the possibility of type I error. Third, body composition was assessed using bioelectrical impedance analysis (BIA) rather than dual-energy X-ray absorptiometry (DXA), which is considered the gold standard and may have introduced measurement bias. Fourth, the intervention period was limited to 12 weeks, which restricts conclusions regarding longer-term effects. Finally, the study population was relatively small and homogeneous, thereby limiting the generalizability of the findings to broader and more diverse populations.
Minor points
- Commercial devices and materials should be cited with full manufacturer information (company, city, country) at first mention.
Response:
Thank you for your valuable comments. We have added the manufacturer information (company, city, country) of the test soymilk and the equipment used in Materials and Methods section as follows:
MARUSAN-AI Co., Ltd., Okazaki, Japan (lines 104-105)
InBody Co., Ltd., Seoul, Korea (lines 112-113)
TANITA Corporation, Tokyo, Japan (line 116)
- The formulas used to calculate UUN (Tanaka) and protein intake (Maloni) are cited only by name. For transparency, the authors could consider briefly describing these formulas.
Response:
Thank you for your helpful suggestions. We have added the following explanation of the calculation formula used in the Methods section as follows (lines 116-121):
Estimated protein intake (EPI) was calculated using the Maloni formula [10]. The Maloni formula assumes that urinary nitrogen accounts for the majority of nitrogen loss and calculates daily protein intake from 24-hour urinary urea nitrogen (UUN) and creatinine values. UUN was calculated using the Tanaka formula [11], which calculates daily UUN excretion from spot urine samples based on urinary urea and creatinine concentrations, and body weight.
- The subgroup thresholds used in the analyses (e.g., ≥5000 steps/day, ≥30 kcal/kg/day, ≥1.0 m/s walking speed) are reasonable, but the manuscript should briefly justify or cite the evidence supporting these cut-offs .
Response:
We appreciate your valuable comments and suggestions to improve our manuscript. We have added the following explanation to the new “2.4. Each cutoff values for subgroups” section regarding the clinical rationale for each cutoff value (line 126 to 152). Additionally, references (Ref. No. 15, 17, 18, 19) have been added to the References section.
2.4. Each cutoff values for subgroups
A decrease in walking speed is one of the criteria for assessing frail (J-CHS criteria), and the cutoff value for walking speed is 1.0 m/s [9].
Step count is one of the popular indicators for assessing physical activity. Based on Tudor-Locke et al.'s classification of daily physical activity levels based on step count, the cutoff value for step count was set at 5,000 steps/day. A step count of less than 5,000 steps/day is considered a "sedentary lifestyle" in which the daily physical activity level is low or high, with little exercise or sports, while a step count of 5,000 steps/day or more is considered a low- to high-level exercise or sports [13].
Physical activity is one of the frailty assessment criteria (J-CHS criteria), and total energy expenditure (TEE) is an indicator of physical activity. Based on age-specific TEE data, the cutoff value was set at 30 kcal/kg body weight/day [14].
Total energy intake (TEI) is strongly associated with the onset and progression of frailty in elderly people, and sufficient intake may have a preventative effect. In particular, low energy intake has been reported to be associated with an increased risk of frailty [15]. Based on data on the estimated energy requirements of elderly Japanese people, the cutoff value was set at 30 kcal/kg body weight/day [16].
Adequate protein intake has been reported to contribute to maintaining muscle mass and preventing frail [17, 18]. To prevent the onset of frail and sarcopenia, it is recommended that Japanese elderly (aged 65 years or older) consume at least 1.0 g/kg body weight/day of protein [19]. Therefore, the cutoff value for estimated protein intake (EPI) was set at 1.0 g/kg body weight/day.
Additional references (lines 389-390, 393-399):
15. Lorenzo-López, L.; Maseda, A.; de Labra, C.; Regueiro-Folgueira, L.; Rodríguez-Villamil, J. L.; Millán-Calenti, J. C. Nutritional determinants of frailty in older adults: A systematic review. BMC Geriatr. 2017, 108.
17. Nanri, H.; Watanabe, D.; Yoshida, T.; Yoshimura, E.; Okabe, Y.; Ono, M.; Koizumi, T.; Kobayashi, H.; Fujita, H.; Kimura, M.; Yamada, Y. Adequate protein intake on comprehensive frailty in older adults: Kyoto-Kameoka study. J. Nutr. Health Aging 2022, 26, 161–168.
18. Mendonça, N.; Kingston, A.; Granic, A.; Jagger, C. Protein intake and transitions between frailty states and to death in very old adults: the Newcastle 85+ study. Age Ageing 2019, 49, 32–38.
19. Ministry of Health, Labour and Welfare. Report of the Study Group on the Formulation of Dietary Intakes for Japanese (2020 Edition) [in Japanese]. 2019, p 114. https://www.mhlw.go.jp/content/10904750/000586553.pdf
- Adding a few recent studies on soy protein and muscle health would strengthen the background.
Response:
Thank you very much for your valuable comments regarding the addition of recent research on soy protein and muscle health. In response, we have added the following reference to recent studies in the Introduction section (lines 60-66):
Previous studies have reported that soy protein intake is effective for maintaining muscle mass and strength across various levels of physical activity, with particularly pronounced effects in individuals with regular exercise habits [3]. Recent research indicates that soy protein offers benefits for muscle adaptation, antioxidant status, and hormonal responses, contributing to improved exercise performance [4]. Furthermore, plant-based proteins, including soy, have been reported to promote recovery from muscle damage and contribute to reduced inflammatory markers [5].
Additional references (lines 357-366):
3. Hashimoto, R.; Sakai, A.; Murayama, M.; Ochi, A.; Abe, T.; Hirasaka, K.; Ohno, A.; Teshima-Kondo, S.; Yanagawa, H.; Yasui, N.; et al. Effects of dietary soy protein on skeletal muscle volume and strength in humans with various physical activities. J. Med. Invest. 2015, 62, 177–183.
4. Zare, R.; Devrim-Lanpir, A.; Guazzotti, S.; Redha, A. A.; Prokopidis, K.; Spadaccini, D.; Cannataro, R.; Cione, E.; Henselmans, M.; Aragon, A. A. Effect of Soy Protein Supplementation on Muscle Adaptations, Metabolic and Antioxidant Status, Hormonal Response, and Exercise Performance of Active Individuals and Athletes: A Systematic Review of Randomised Controlled Trials. Sports Med. 2023, 53, 2417–2446.
5. Govindasamy, K.; Parpa, K.; Katanic, B.; Clark, C. C. T.; Elayaraja, M.; Kambitta Valappil, I. N.; Dulceanu, C.; Geantă, V. A.; Tolan, G. A.; Zouhal, H. Effect of Plant-Based Proteins on Recovery from Resistance Exercise-Induced Muscle Damage in Healthy Young Adults—A Systematic Review. Nutrients 2025, 17, 2571.

Round 2
Reviewer 1 Report
Comments and Suggestions for Authors
This revised manuscript has a great improvement and the reviewer has no further comments.
Author Response
Thank you very much for your support!
Best regards!
Reviewer 2 Report
Comments and Suggestions for Authors
Article is improved but not fully corrected:
-“mean difference in effect size” correct to “mean difference in SMI change”
-“Analysis of covariance with age, sex, and baseline values as covariates.” – incomplete sentence
-whether cutoffs were prespecified or purely exploratory and why they matter physiologically for muscle mass outcomes
Comments on the Quality of English Language-not clear table description sStudent, AAnalysis
-fix grammar issues like: “Previous our study has also shown…”
Author Response
Article is improved but not fully corrected:
- -“mean difference in effect size” correct to “mean difference in SMI change”
Response:
We appreciate your valuable comments. As you pointed out, we have corrected "mean difference in effect size" to "mean difference in SMI change." (line 33 and 182)
- -“Analysis of covariance with age, sex, and baseline values as covariates.” – incomplete sentence
Response:
Thank you very much for your careful reading and we really appreciate and agree with your comments. As you pointed out, we have corrected as follows (lines 155-157):
An analysis of covariance was conducted to evaluate differences in SMI change, adjusting for age, sex, and baseline values.
- -whether cutoffs were prespecified or purely exploratory and why they matter physiologically for muscle mass outcomes
Response:
We appreciate your careful comments. The cutoff values were set exploratory and post hoc. The physiological significance of the cutoff values for muscle mass outcomes is described in section 2.4. Each cutoff value for subgroups (lines 127-148).
In particular, adequate protein intake has been reported to contribute to the maintenance of muscle mass and the prevention of frailty [17, 18]. To prevent the onset of frailty and sarcopenia, it is recommended that Japanese elderly individuals (aged 65 years or older) consume at least 1.0 g/kg body weight/day of protein [19]. TEI is strongly associated with the onset and progression of frailty in elderly individuals, and sufficient intake may have a preventative effect. In particular, low energy intake has been reported to be associated with an increased risk of frailty [15]. Based on data on the estimated energy requirements of elderly Japanese individuals, the cutoff value was set at 30 kcal/kg body weight/day [16].

Reviewer 3 Report
Comments and Suggestions for Authors
Thank you very much for your careful and thorough revision of the manuscript. I sincerely appreciate the detailed and thoughtful way in which you have addressed each of the points raised in my review.
The improvements in methodological clarity, the expanded discussion, and the transparent acknowledgment of the study’s limitations have significantly strengthened the quality and credibility of the work. I would like to congratulate you on this rigorous and well-executed revision.
Author Response

(The authors gave the same response as above.)
